# ENHANCING LOGICAL REASONING OF LARGE LANGUAGE MODELS VIA PHASED FINE-TUNING

## ABSTRACT

Large language models (LLMs) have not only achieved impressive progress in natural language processing tasks but also demonstrated remarkable performance in practical applications such as intelligent customer service. However, LLMs continue to demonstrate shortcomings in complex logical reasoning and decision-making capabilities. As one of the key elements in human intelligence, logical reasoning plays a crucial role in various tasks, including natural language understanding, intelligent question-answering, and knowledge graph construction. The deficiency of LLMs in logical reasoning significantly limits their applications, especially in domains requiring high accuracy and trustworthiness. To tackle this issue, we focus on propositional logic and introduce a logic QA-specific phased fine-tuning method to enhance the logical reasoning capabilities of LLMs, performing supervised fine-tuning from easy to hard. Based on the symbolic logical form complexity derived from disjunctive normal form (DNF) and the LLM logical reasoning complexity in propositional logic question-answering tasks, the difficulty of logic question-answering samples is automatically computed, and the training data is stratified based on the difficulty. A dedicated fine-tuning dataset for propositional logic is constructed, and experiments demonstrate the significant effectiveness of our method, especially in those tasks demanding strong logical reasoning ability.

## 1 INTRODUCTION

Large language models (LLMs) have demonstrated impressive performance across a wide range of natural language processing tasks. However, recent studies have found that there are still significant challenges regarding the logical reasoning abilities of LLMs (Luo et al., 2023; Calanzone et al., 2025). Logical reasoning tasks require LLMs to perform complex deductive, inductive, or abductive reasoning over a set of premises and logical rules (Morishita et al., 2024; Sun et al., 2024). Specifically, models are asked to determine whether a given conclusion can be logically inferred from the provided premises. For instance, given premises such as "*All mammals are animals; all dogs are mammals; all cats are mammals; all Golden Retrievers are dogs*", the LLM is required to determine the statement "*All Golden Retrievers are animals*" from the choices "*True, False or Unknown*". To answer correctly, the model has to construct the logical reasoning chain "*Golden Retrievers → dogs → mammals → animals*" and thus infer the statement is *True*. When facing problems that cannot be directly solved by pattern matching from common-sense knowledge in the training set, the complex logical reasoning ability is crucial to guarantee a human-like cognitive intelligence (Mirzadeh et al., 2025).

Surprisingly, the LLaMA-13B model achieves only 33.63% accuracy under 8-shot prompting on FOLIO (Han et al., 2024), barely outperforming random guessing (33.33%), which highlights that pre-trained LLMs with in-context learning fail to solve complex logical reasoning tasks. Numerous efforts have been made to enhance the logical reasoning capabilities of LLMs, which can be generally classified into three categories. Solver-based methods firstly translate natural language problems into symbolic logic representations, and then leverage an external logic solver to infer answers (Ye et al., 2023; Pan et al., 2023; Olausson et al., 2023). Prompt-based methods fall into two main paradigms: one prompts LLMs to generate explicit logical chains in natural language (Wei et al., 2022; Yao et al., 2024; Zhang et al., 2024), while the other uses LLMs to translate into symbolic language, logically infer answers and verification via carefully designed prompts (Xu et al., 2024; Liu et al., 2024; Li et al., 2024b). Fine-tuning approaches enhance logical reasoning performance by either constructing synthetic datasets with detailed logical deduction steps (Bao et al., 2024; Morishita et al., 2024) or

Figure 1: (a) Overview of the phased fine-tuning framework for enhancing the LLM logical reasoning ability; (b) Proposed framework for computing symbolic logical form complexity and LLM logical reasoning complexity based on the disjunctive normal form (DNF) transformation.

augmenting training data with logical reasoning examples (Feng et al., 2024; Wan et al., 2024; Jiao et al., 2024), thereby improving both model accuracy and interpretability in logical reasoning tasks.

However, solver-based approaches are prone to low execution rates due to translation errors and face problems such as search explosion for complex logical reasoning problems (Liu et al., 2024; Li et al., 2024b). Prompt-based approaches rely on the original logical reasoning abilities of pre-trained models, yet empirically, LLMs such as LLaMA-13B with in-context learning have been shown to perform only marginally better than random guessing in logical QA tasks (Yang et al., 2024a). In addition, we found that directly fine-tuning the LLMs is unable to effectively solve complex logical reasoning problems. This is because logical reasoning problems contain complex logical rules implicitly compared to general reasoning problems (Luo et al., 2023), as well as the inability of LLMs to comprehend and integrate multiple complex logical premises and logical rules directly to infer the truth of the conclusions (Morishita et al., 2024). These significantly restrict the practical application of large language models in scenarios such as intelligent question-answering and autonomous decision-making. Therefore, there still needs to develop more effective fine-tuning paradigms for the logical reasoning of LLMs.

To fill this gap, we focus on propositional logic QA within logical reasoning tasks in this study and propose a novel phased fine-tuning strategy to enhance LLMs' logical reasoning by a curriculum learning (Bengio et al., 2009) framework that progressively trains LLM from easy to difficult tasks, as Figure 1 shows. The most important step in curriculum learning lies in the difficulty measurement of logical QA tasks, which has not been studied before. In this paper, we adhere to the nature of logical QA and define the difficulty of solving a logical QA task as the complexity of determining a proposition's truth value, which depends on two factors: (a) the symbolic logical form complexity based on a partial order of disjunctive normal form (DNF) and (b) the LLM logical reasoning complexity based on the probabilities of atomic propositions in DNF being true. This forms the core idea of our propositional logic difficulty measurement method in this study.

The contributions of this paper include four main aspects:

- We propose a propositional logic difficulty decomposition method based on syntactic complexity and semantic understanding difficulty, which evaluates problem difficulty by convert-

ing natural language logical problems into standardized Disjunctive Normal Form (DNF) and considering both logical form complexity and LLM logical reasoning complexity.

- We design a phased fine-tuning framework for propositional logic QA, organizing training data from easy to hard to enable incremental gain of logical reasoning ability.

- We construct a propositional logic-specific fine-tuning dataset, providing a valuable benchmark through symbolic translation and detailed difficulty annotation.

- Extensive experiments validate the effectiveness of our method, showing that phased fine-tuning significantly improves model performance on standard logical reasoning benchmarks as well as general reasoning tasks, especially in handling complex, long reasoning steps. This work not only provides an effective path for enhancing LLMs' logical reasoning ability but also offers a scalable approach for more complex logical reasoning tasks.

## 2 PRELIMINARIES

### 2.1 PROPOSITIONAL LOGIC

Propositional logic represents propositions in natural language and the logical relations among them through formal symbolic notation. Each proposition is recursively constructed from atomic propositions and logical connectives. An atomic proposition is a statement that cannot be further decomposed into sub-propositions. Logical connectives express the logical relationships between propositions, and the corresponding meanings in natural language of logical connectives can be found in the supplementary material. These logical connectives cover the primary types of relations between propositions or sentences. To determine which forms of inference are valid, logic introduces truth-value semantics, which assigns a truth value to each atomic proposition via a truth-value function. The truth-value function is defined as $Pron \rightarrow V$, where $Pron$ denotes the set of all atomic propositions and $V = \{T, F\}$ represents the range of truth values, with $T$ and $F$ indicating "true" and "false," respectively. Based on this, the truth values of compound propositions formed by logical connectives can be inductively defined as shown in the supplementary material.

### 2.2 LOGICAL QUESTION ANSWERING IN NATURAL LANGUAGE

In the logical question answering task, we have $m$ logic QA samples of the form $\{P_i^{(1)}, P_i^{(2)}, \ldots, P_i^{(n_i)}, C_i, Y_i\}_{i=1}^m$, where $P_i^{(1)}, P_i^{(2)}, \ldots, P_i^{(n_i)}$ are the premises in natural language form, $C_i$ is the conclusion in natural language form, and $Y_i$ is the label indicating whether $C_i$ is true given the premises. If the conclusion can be logically derived from the premises, it is labeled as "true"; if the conclusion can be proved to be false based on the premises, it is labeled as "false"; and if the conclusion's truth value cannot be determined from the premises, the output should be "undetermined/unknown."

The following is an example of a logic QA task. The premises given to the model are: *"Premise 1: If Tom reads "Game Theory", then he will gain knowledge", "Premise 2: If Tom gains knowledge, then he will become smarter", and "Premise 3: Tom reads "Game Theory".* The question is then posed: *"Is the following assertion true: Tom will become smarter?"* The correct answer should be *"True"*, because based on the three premises and the rules of inference, we can derive the conclusion *"Tom gains knowledge"*, which infers the assertion *"Tom will become smarter"*.

It is important to note that, since logic QA tasks are designed to assess the reasoning ability of large models, the given premises are typically assumed to be true, or consistent with factual knowledge or a knowledge base. This contrasts with reasoning in propositional logic, which also considers cases where the premises may be false. In propositional logic, if the premises are false, any conclusion can be inferred as true, meaning the inference is always valid. However, in logic QA tasks for LLMs, such scenarios where the premises are false are generally not considered.

## 3 METHOD

In this study, we propose a method for measuring the difficulty of logical QA by combining the complexity in symbolic language of propositional logic, along with the complexity in natural language.

The former captures the difficulty of reasoning from premises to conclusion using logical symbolic language, while the latter reflects the difficulty for a pretrained LLM to understand the premises and conclusion in natural language. This idea aligns with the principle in logic that "the validity of a conclusion depends on the truth of the premises and the validity of the argument." Based on this difficulty measurement, we propose a phased fine-tuning framework that stratifies samples in logic QA data and fine-tunes LLMs from easy to hard to achieve better reasoning ability.

### 3.1 TRANSLATION FROM NATURAL LANGUAGE TO SYMBOLIC LANGUAGE

It is important to note that each premise and conclusion may consist of multiple atomic propositions, so their complexity cannot simply be measured by the number of premises $n_i$. To assess the difficulty of the propositional logic QA task, we need to translate the natural language into symbolic language. We leverage a pretrained LLM to perform this translation.

The translation involves two steps. First, we extract atomic propositions from the set of premises and assign symbolic representations, resulting in a mapping between symbols and natural language expressions. After obtaining the mapping, we then represent the original premises and conclusion using symbolic language. Since atomic propositions may be expressed differently across different premises in natural language, we again utilize the large language model to perform this conversion. See the supplementary material for an example of the prompts and outputs used in this step.

### 3.2 SYMBOLIC LOGICAL FORM COMPLEXITY

After translation, the natural language logic QA data is converted into symbolic language. We denote the resulting symbolic data as $\{P_i^{(1)}, P_i^{(2)}, \ldots, P_i^{(n_i)}, C_i, Y_i\}_{i=1}^m$, where each lowercase symbol represents a formula of a proposition, possibly composed of multiple atomic propositions. The logic QA task is equivalent to determining the truth value of the following formula:

$$\left( \bigwedge_{l=1}^{n_i} P_i^{(l)} \right) \to C_i. \tag{1}$$

Although we have expressed the problem in symbolic form, due to the equivalence of logical formulas, the same semantic content can be expressed using different syntactic structures. For example, the formulas $P \to Q$ and $\neg P \vee Q$ are logically equivalent, although the former is an implication and the latter is a disjunction. As a more complex example, $(P \wedge Q) \vee (R \wedge S)$ is logically equivalent to $(P \vee R) \wedge (P \vee S) \wedge (Q \vee R) \wedge (Q \vee S)$. Clearly, equivalent formulas can vary significantly in form, and thus, the complexity of symbolic language cannot be reliably measured directly from the original translated formulas. Instead, we must transform all original formulas into a unified and comparable form. In propositional logic, canonical forms such as disjunctive normal form (DNF) possess a well-defined and consistent syntactic structure.

**Definition 3.1.** *A disjunctive normal form (DNF) is a disjunction of conjunctions, where each conjunction consists of atomic propositions or their negations. It is formally expressed as:*

$$Q = \bigvee_{i=1}^{n} \left( \bigwedge_{j=1}^{m_i} E_{ij} \right), \tag{2}$$

*where $E_{ij}$ is an atomic proposition or the negation of an atomic proposition.*

Classical results in logic, such as the DNF theorem, state that every propositional formula can be equivalently transformed into DNF. This provides the foundation for defining a unified measure of complexity in symbolic language for propositional logic formulas. We propose that the total length of a DNF formula $\sum_{i=1}^{n} m_i$ can serve as a measurement for its complexity. In the following, we elaborate on the rationale behind this measurement.

**Lemma 3.1.** *Every propositional logic formula can be transformed into an equivalent disjunctive normal form (DNF) (Davey & Priestley, 2002).*

The dual form of DNF is conjunctive normal form (CNF). Theoretically, there is a similar result for CNF as Lemma 3.1, and CNF can also serve as a standard form in complexity measurement.

However, the logic QA tasks are all of the form $(\bigwedge_{l=1}^{n_i} P_l^i) \to C_i$, transforming them into CNF often leads to an exponential explosion, whereas the resulting DNF is typically more compact. Empirically, on some samples with an average number of premises of 8.2, we find that the average transformed DNF length is 19.2, while the average CNF length is 9568.5. When converted to DNF, the resulting formula lengths are inflated on average by a factor of 2.53 compared to the number of prerequisites, while for CNF, the result is 577.85. See the supplementary material for more details.

When measuring syntactic complexity, since all formulas have been transformed into the form in Definition 3.1, we can ignore the differences between atomic propositions themselves and focus only on quantitative characteristics. The syntactic complexity of each DNF formula $\bigvee_{i=1}^{n} \left( \bigwedge_{j=1}^{m_i} E_{ij} \right)$ is entirely determined by a sequence of integers $(m_1, m_2, \ldots, m_n)$. Without loss of generality, we assume $m_1 \geq m_2 \geq \ldots \geq m_n$; otherwise, we reorder the DNF formula using the commutative law. Given two DNF formulas $Q$ and $Q'$ with corresponding sequences $(m_1, m_2, \ldots, m_n)$ and $(m_1', m_2', \ldots, m_{n'}')$, we define a binary relation $\succ$ between these two sequences as follows:

**Definition 3.2.** $(m_1, \ldots, m_n) \succ (m_1', \ldots, m_{n'}') \iff n \geq n', \ m_1 \geq m_1', \ \ldots, \ m_{n'} \geq m_{n'}'$.

In fact, $(m_1, m_2, \ldots, m_n) \succ (m_1', m_2', \ldots, m_{n'}')$ indicates that the DNF formula corresponding to the former is **strictly** more difficult than the latter in form, because it represents a disjunction of more conjunctive clauses, and its largest conjunctive clause contains more atomic propositions. If neither $(m_1, m_2, \ldots, m_n) \succ (m_1', m_2', \ldots, m_{n'}')$ nor $(m_1', m_2', \ldots, m_{n'}') \succ (m_1, m_2, \ldots, m_n)$ holds, we consider the two formulas incomparable in difficulty: one contains more disjunctions, while the other contains more atomic propositions in the largest conjunction. Let $S$ be the set of integer sequences corresponding to all DNF formulas transformed from the dataset. Then $\succ$ is a partial order on $S$. The proof for the following proposition is in the supplementary material.

**Proposition 3.1.** $\succ$ *defines a partial order on the set* $S$.

We aim to define a complexity measurement function $\mathcal{C}_{SL} : S \to \mathbb{R}_+$, such that samples that are strictly more difficult receive a higher complexity score. This gives rise to the following **Difficulty Partial Order Principle**.

**Definition 3.3.** *We say a complexity measurement function satisfies the Difficulty Partial Order Principle if any* $(m_1, m_2, \ldots, m_n) \succ (m_1', m_2', \ldots, m_{n'}')$*, we have*

$$\mathcal{C}_{SL}((m_1, m_2, \ldots, m_n)) \geq \mathcal{C}_{SL}((m_1', m_2', \ldots, m_{n'}')).$$

We can prove that using the DNF length as the definition of complexity in symbolic language, i.e., $\mathcal{C}_{SL}((m_1, m_2, \ldots, m_n)) = \sum_{i=1}^{n} m_i$ satisfies the above Difficulty Partial Order Principle.

**Theorem 3.1.** $\mathcal{C}_{SL}((m_1, m_2, \ldots, m_n)) := \sum_{i=1}^{n} m_i$ *satisfies the Difficulty Partial Order Principle.*

*Proof.* If $(m_1, m_2, \ldots, m_n) \succ (m_1', m_2', \ldots, m_{n'}')$, then we have $n \geq n', \ m_1 \geq m_1', \ \ldots, \ m_{n'} \geq m_{n'}'$. Therefore,

$$\mathcal{C}_{SL}((m_1, m_2, \ldots, m_n)) = \sum_{i=1}^{n} m_i \geq \sum_{i=1}^{n'} m_i \geq \sum_{i=1}^{n'} m_i' = \mathcal{C}_{SL}((m_1', m_2', \ldots, m_{n'}')).$$

$\square$

## 3.3 LLM Logical Reasoning Complexity

In original logical QA tasks, both premises and conclusions are in natural language form. However, the aforementioned complexity in symbolic language only considers the form of logical formulas, without accounting for the complexity in natural language.

Based on the transformed DNF as Eq. 3.1, Let $Pr_w(P = \top)$ be the probability of the proposition $P$ being true, given by the LLM we would like to fine-tune. Since we have extracted the atomic proposition of the smallest unit from the premises, we follow recent neuro-symbolic AI work (van Krieken et al., 2024; Calanzone et al., 2025) to make the following assumption of independence.

**Assumption 3.1.** *Given the parameters of the LLM, the atomic propositions $E_{ij}$ are independent from each other. Formally $E_{11} \perp\!\!\!\perp E_{12} \perp\!\!\!\perp \cdots \perp\!\!\!\perp E_n^{m_n} | w$.*

Note that we only rely on the independence between atomic propositions instead of the premises, posing a weaker assumption. Then we can compute the probability of the logic QA being true as follows:

$$\text{Pr}_\text{w}(Q = \top) = 1 - \prod_{i=1}^{n}\left(1 - \prod_{j=1}^{m_i}\text{Pr}_\text{w}(E_{ij} = \top)\right),$$

. We then defined the LLM logical reasoning complexity based on the following information entropy:

$$\mathcal{C}_{NL} = -\text{Pr}_\text{w}(Q = \top)\log_2(\text{Pr}_\text{w}(Q = \top)) - (1 - \text{Pr}_\text{w}(Q = \top))\log_2(1 - \text{Pr}_\text{w}(Q = \top)).$$

For the example in Figure 1, if we have $\text{Pr}_w(P = \top) = 0.2$, $\text{Pr}_w(Q = \top) = 0.4$ and $\text{Pr}_w(C = \top) = 0.7$, then the probability of the original logical QA being true can be computed as:

$$\text{Pr}_\text{w}\left((P \wedge \neg Q) \vee (Q \wedge \neg C) \vee \neg P \vee C = \top\right)$$
$$=1 - (1 - \text{Pr}_\text{w}(P = \top)\text{Pr}_\text{w}(Q =\bot))(1 - \text{Pr}_\text{w}(Q = \top)\text{Pr}_\text{w}(C =\bot))(1 - \text{Pr}_\text{w}(P =\bot))(1 - \text{Pr}_\text{w}(C = \top))$$
$$=1 - (1 - 0.2 * 0.6)(1 - 0.4 * 0.3)(1 - 0.8)(1 - 0.7) = 0.95.$$

Note that the conclusion in the original QA problem was $C$, and that the probability that the LLM alone will determine whether $C$ is correct is different from the result from the above computation. This reflects the difference between the logical QA task and general QA.

## 3.4 DATASET STRATIFICATION BASED ON DIFFICULTY

For a logical QA sample, we have obtained both the symbolic logical form complexity and the LLM logical reasoning complexity. These need to be combined into an overall sample difficulty measure for data stratification. We find that the empirical distributions of both complexities are close to normal, which can be seen in the supplementary material.

Let $S_i = (P_i^{(1)}, P_i^{(2)}, \ldots, P_i^{(n_i)}, C_i)$ be sample $i$ in natural language, and $s_i$ be the corresponding sequence of integer in its DNF. We define the final sample complexity as follows:

$$\mathcal{C}_{final,i} = \alpha\mathcal{C}_{SL,i} + (1 - \alpha)\mathcal{C}_{NL,i}, \tag{3}$$

where $\mathcal{C}_{SL,i}$ is the normalized value of the formal complexity $\mathcal{C}_{SL}(s_i)$ for sample $i$, i.e., $\frac{\mathcal{C}_{SL}(s_i) - \min_k \mathcal{C}_{SL}(s_k)}{\max_k \mathcal{C}_{SL}(s_k) - \min_k \mathcal{C}_{SL}(s_k)}$, and $\mathcal{C}_{NL,i}$ is the normalized value of the natural language complexity for sample $i$ similarly.

We observe that the final sample difficulty is approximately normally distributed, with a maximum around 0.7, indicating that the rankings of formal and natural language complexities are not entirely consistent. See more details in the supplementary material. We stratify the training data into three parts with increasing difficulty, and fine-tune LLM on these parts sequentially in a phased manner. In other words, instead of training with the randomly shuffled whole dataset, we split the dataset according to the difficulty measurement and only allow the shuffle within the same subparts.

## 4 THEORETICAL ANALYSIS

Let $\mathcal{D}$ be the distribution of training samples $\{S_t\}_{t=1}^{n}$ (suppose the index corresponds to the training order). Let w be the tunable parameter vector of LLM. Let $L(S_t, \text{w})$ denote the loss of the model with parameter w when given $S_t$. We assume the last layer of LLM is tunable, under the objective:

$$\min_\text{w} L_\mathcal{D}(\text{w}) = \mathbb{E}_{S_t \sim \mathcal{D}}(L(S_t, \text{w})).$$

SGD computes a sequence of estimators $\{\text{w}_t\}_{t=1}^{n}$. Although in practice, a variant of SGD with batch size larger than 1 is used, we analyze here a basic form with the update rule:

$$\text{w}_{t+1} = \text{w}_t - \eta\frac{\partial L(S_t, \text{w})}{\partial \text{w}}|_{\text{w}=\text{w}_t},$$

where $\eta$ is the learning rate. Let $\bar{\text{w}} = \arg\min_\text{w} L_\mathcal{D}(\text{w})$ be the optimal parameter vector, based on which we can define the *Difficulty Score* as:

$$\Psi(S_t) = L(S_t, \bar{\text{w}}).$$

Let $\Delta_t(\Psi) = \mathbb{E}[||\text{w}_t - \bar{\text{w}}||^2 - ||\text{w}_{t+1} - \bar{\text{w}}||^2 \mid \Psi]$ be the expected convergence rate at step $t$, given fixed difficulty score $\Psi$. Then we have the following result, motivating training easiest samples first.

Table 1: Comparison of our phased instruction fine-tuning and one-off instruction fine-tuning baselines across different reasoning benchmark datasets. RI denotes *relative improvement* over the corresponding baseline.

| Benchmark | Metric | LLaMA | LLaMA-PFT | RI | Ministral | Ministral-PFT | RI |
|---|---|---|---|---|---|---|---|
| MMLU | acc | $0.608 \pm 0.003$ | $\mathbf{0.614 \pm 0.003}$ | 0.986% ↑ | $0.489 \pm 0.004$ | $\mathbf{0.506 \pm 0.004}$ | 3.497% ↑ |
| BIG-bench | exact_match | $0.297 \pm 0.005$ | $\mathbf{0.330 \pm 0.005}$ | 10.992% ↑ | $0.101 \pm 0.003$ | $\mathbf{0.116 \pm 0.003}$ | 14.680% ↑ |
| RobustLR | acc | $0.690 \pm 0.008$ | $\mathbf{0.730 \pm 0.008}$ | 5.791% ↑ | $0.624 \pm 0.008$ | $\mathbf{0.629 \pm 0.008}$ | 0.753% ↑ |
| LogicNLI | acc | $0.355 \pm 0.010$ | $\mathbf{0.375 \pm 0.010}$ | 5.775% ↑ | $0.414 \pm 0.011$ | $\mathbf{0.440 \pm 0.011}$ | 6.280% ↑ |
| SNLI | acc | $0.536 \pm 0.005$ | $\mathbf{0.639 \pm 0.004}$ | 19.321% ↑ | $0.480 \pm 0.005$ | $\mathbf{0.485 \pm 0.005}$ | 1.187% ↑ |
| RTE | acc | $0.783 \pm 0.008$ | $\mathbf{0.833 \pm 0.007}$ | 6.410% ↑ | $0.815 \pm 0.007$ | $\mathbf{0.820 \pm 0.007}$ | 0.588% ↑ |
| ARC (challenge) | acc | $0.514 \pm 0.014$ | $\mathbf{0.533 \pm 0.014}$ | 3.654% ↑ | $0.469 \pm 0.014$ | $\mathbf{0.471 \pm 0.014}$ | 0.533% ↑ |
| MathQA | acc_norm | $0.380 \pm 0.008$ | $\mathbf{0.398 \pm 0.009}$ | 4.572% ↑ | $0.338 \pm 0.008$ | $\mathbf{0.341 \pm 0.008}$ | 0.797% ↑ |
| FOLIO | exact_match | $0.410 \pm 0.015$ | $\mathbf{0.448 \pm 0.015}$ | 9.255% ↑ | $\mathbf{0.504 \pm 0.015}$ | $0.503 \pm 0.015$ | 0.199% ↓ |
| LogiQA2.0 | acc_norm | $0.363 \pm 0.012$ | $\mathbf{0.386 \pm 0.012}$ | 6.305% ↑ | $0.298 \pm 0.011$ | $\mathbf{0.306 \pm 0.011}$ | 2.782% ↑ |

Table 2: Ablation study results on different benchmarks with and without $C_{SL}$ and $C_{NL}$ for LLaMA and Minstral.

| Benchmark | Metric | LLaMA-PFT | w/o $C_{SL}$ | w/o $C_{NL}$ | Ministral-PFT | w/o $C_{SL}$ | w/o $C_{NL}$ |
|---|---|---|---|---|---|---|---|
| MMLU | acc | $\mathbf{0.614 \pm 0.003}$ | $0.533 \pm 0.004$ | $0.527 \pm 0.004$ | $\mathbf{0.506 \pm 0.004}$ | $0.505 \pm 0.004$ | $0.463 \pm 0.004$ |
| Big-bench | exact_match | $\mathbf{0.330 \pm 0.005}$ | $0.154 \pm 0.004$ | $0.145 \pm 0.004$ | $\mathbf{0.116 \pm 0.003}$ | $0.054 \pm 0.002$ | $0.084 \pm 0.003$ |
| RobustLR | acc | $\mathbf{0.730 \pm 0.008}$ | $0.591 \pm 0.009$ | $0.624 \pm 0.008$ | $\mathbf{0.629 \pm 0.008}$ | $0.543 \pm 0.009$ | $0.583 \pm 0.009$ |
| LogicNLI | acc | $\mathbf{0.375 \pm 0.010}$ | $0.333 \pm 0.010$ | $0.359 \pm 0.010$ | $\mathbf{0.440 \pm 0.011}$ | $0.358 \pm 0.010$ | $0.358 \pm 0.010$ |
| SNLI | acc | $\mathbf{0.639 \pm 0.004}$ | $0.566 \pm 0.005$ | $0.494 \pm 0.005$ | $\mathbf{0.485 \pm 0.005}$ | $0.456 \pm 0.005$ | $0.298 \pm 0.004$ |
| RTE | acc | $\mathbf{0.833 \pm 0.007}$ | $0.777 \pm 0.008$ | $0.798 \pm 0.008$ | $\mathbf{0.820 \pm 0.007}$ | $0.781 \pm 0.008$ | $0.807 \pm 0.007$ |
| ARC (challenge) | acc | $\mathbf{0.533 \pm 0.014}$ | $0.502 \pm 0.014$ | $0.495 \pm 0.014$ | $\mathbf{0.471 \pm 0.014}$ | $0.460 \pm 0.014$ | $0.447 \pm 0.014$ |
| MathQA | acc_norm | $\mathbf{0.398 \pm 0.009}$ | $0.375 \pm 0.008$ | $0.369 \pm 0.008$ | $\mathbf{0.341 \pm 0.008}$ | $0.340 \pm 0.008$ | $0.337 \pm 0.008$ |
| FOLIO | exact_match | $\mathbf{0.448 \pm 0.015}$ | $0.284 \pm 0.014$ | $0.335 \pm 0.014$ | $\mathbf{0.503 \pm 0.015}$ | $0.489 \pm 0.015$ | $0.499 \pm 0.015$ |
| LogiQA2.0 | acc_norm | $\mathbf{0.386 \pm 0.012}$ | $0.321 \pm 0.011$ | $0.307 \pm 0.011$ | $0.306 \pm 0.011$ | $0.297 \pm 0.011$ | $\mathbf{0.307 \pm 0.011}$ |

**Theorem 4.1.** *At step $t$, the expected convergence rate for training sample $S_t$ is monotonically decreasing with the Difficulty Score $\Psi(S_t)$. Formally, we have*

$$\frac{\partial \Delta_t(\Psi)}{\partial \Psi} = -8\eta^2 \mathbb{E}_{S_t \sim \mathcal{D}|\Psi} \left[ \|S_t\|^2 \right] \Psi \leq 0.$$

**Corollary 4.2.** *To achieve faster convergence to $\bar{w}$, we should choose the easier sample at each step, thus resulting in training from easy to hard.*

The above result demonstrates that training with easier samples achieves faster convergence, which motivates our phased fine-tuning based on the sample's logical complexity. See supplementary material for the detailed proof.

## 5 EXPERIMENTS

This section aims to validate the effectiveness of our proposed difficulty measurement method for natural language propositional logic QA by comparing the performance of two training strategies: one-off instruction fine-tuning and our proposed phased instruction fine-tuning. All fine-tuning experiments are conducted on the open-source models `LLaMA-3.1 8B-Instruct` (Dubey et al., 2024) and `Ministral-8B-Instruct-2410` (Min, 2024).

**Datasets and Benchmarks.** We construct the training set for propositional logic QA based on the public dataset FLDx2 (Morishita et al., 2024). The 31,420 QA pairs related to propositional logic are our training data. The benchmarks include: FOLIO (Han et al., 2024), SNLI (Bowman et al., 2015), MMLU (Hendrycks et al., 2021), Big-bench (Srivastava et al.) RobustLR (Sanyal et al., 2022), LogicNLI (Tian et al., 2021), RTE datasets (Dagan et al., 2005; Giampiccolo et al., 2007; Bentivogli et al., 2009) , ARC challenge dataset (Clark et al., 2018), MathQA (Amini et al., 2019), LogiQA2.0 (Liu et al., 2023). Datasets annotated with "mcq" denote multiple-choice tasks, and those without annotations are treated as QA tasks by default. For all benchmarks, we evaluate using in-context learning with zero-shot or few-shot settings, which can be seen in the supplementary materials. The reported metrics include accuracy (acc), normalized accuracy over multiple correct answers (acc_norm), and exact match rate (exact_match), which requires character-level exact correspondence.

Table 3: Performance of LLaMA under different instruction training orders. Background cell colors range from light to dark, indicating increasing values within each row.

| Benchmark | Metric | 1-2-3 | 2-1-3 | 3-1-2 | 1-3-2 | 2-3-1 | 3-2-1 |
|---|---|---|---|---|---|---|---|
| MMLU | acc | $0.614 \pm 0.003$ | $0.598 \pm 0.003$ | $\mathbf{0.617 \pm 0.003}$ | $0.611 \pm 0.003$ | $0.612 \pm 0.003$ | $0.600 \pm 0.003$ |
| Big-bench | exact_match | $\mathbf{0.330 \pm 0.005}$ | $0.313 \pm 0.005$ | $0.272 \pm 0.005$ | $0.287 \pm 0.005$ | $0.287 \pm 0.005$ | $0.317 \pm 0.005$ |
| RobustLR | acc | $\mathbf{0.730 \pm 0.008}$ | $0.672 \pm 0.008$ | $0.674 \pm 0.008$ | $0.718 \pm 0.008$ | $0.641 \pm 0.008$ | $0.594 \pm 0.009$ |
| LogicNLI | acc | $0.375 \pm 0.010$ | $0.409 \pm 0.011$ | $\mathbf{0.451 \pm 0.011}$ | $0.424 \pm 0.011$ | $0.418 \pm 0.011$ | $0.342 \pm 0.010$ |
| SNLI | acc | $\mathbf{0.639 \pm 0.004}$ | $0.623 \pm 0.004$ | $0.498 \pm 0.005$ | $0.488 \pm 0.005$ | $0.613 \pm 0.004$ | $0.578 \pm 0.004$ |
| RTE | acc | $\mathbf{0.833 \pm 0.007}$ | $0.827 \pm 0.007$ | $0.810 \pm 0.007$ | $0.784 \pm 0.008$ | $0.749 \pm 0.008$ | $0.806 \pm 0.007$ |
| ARC (challenge) | acc | $0.533 \pm 0.014$ | $0.511 \pm 0.014$ | $\mathbf{0.535 \pm 0.014}$ | $0.513 \pm 0.014$ | $0.531 \pm 0.014$ | $0.517 \pm 0.014$ |
| MathQA | acc_norm | $\mathbf{0.398 \pm 0.009}$ | $0.368 \pm 0.008$ | $0.351 \pm 0.008$ | $0.388 \pm 0.008$ | $0.367 \pm 0.008$ | $0.367 \pm 0.008$ |
| FOLIO | exact_match | $\mathbf{0.448 \pm 0.015}$ | $0.349 \pm 0.015$ | $0.412 \pm 0.015$ | $0.441 \pm 0.015$ | $0.421 \pm 0.015$ | $0.402 \pm 0.015$ |
| LogiQA2.0 | acc_norm | $\mathbf{0.386 \pm 0.012}$ | $0.369 \pm 0.012$ | $0.374 \pm 0.012$ | $0.380 \pm 0.012$ | $0.381 \pm 0.012$ | $0.379 \pm 0.012$ |

**Experimental Setup.** For training, we use LLaMA-Factory (Zheng et al., 2024), and employ DeepSpeed's Zero3 for parallel, full-parameter fine-tuning. All experiments use a consistent learning rate of 5e-6 and 1 training epoch. We set the per-device batch size to 8, gradient accumulation steps to 1, weight decay to 0.1, and warmup ratio to 0.1. A cosine learning rate scheduler is used, following standard fine-tuning practices for large language models.

**Performance Comparison.** We compare the performance of our proposed staged instruction tuning with the baseline one-off instruction fine-tuning across multiple benchmark datasets, as shown in Table 1. Our method significantly outperforms the baseline on nearly all benchmarks. It is worth noting that LLaMA generally outperforms Ministral on these benchmarks. And on this basis, our method still exhibits a more significant performance improvement in the LLaMA model.

**Ablation Study.** We conduct ablation experiments to validate the effectiveness of complexity in symbolic language and complexity in natural language in our proposed approach. Table 2 shows the results on numerous benchmarks. Removing either complexity in symbolic language (w/o $\mathcal{C}_{SL}$) or complexity in natural language (w/o $\mathcal{C}_{NL}$), and performing the same difficulty partitioning and phased fine-tuning, different degrees of performance degradation are observed. On the majority of benchmarks, the loss in performance compared to the baseline is comparable for both ablated versions, but on the natural language inference benchmark SNLI, removing complexity in natural language impairs performance more significantly.

**Phased Fine-tuning for All Permutations of Training Order.** We investigate the effect of training order on the performance of LLaMA on all 10 datasets, as shown in Table 3. Note that the normal order (1-2-3) achieves the overall best performance, which shows it can help LLM to build a basic understanding and confidence, gradually ramping up to harder tasks. In addition, the inverse order (3-2-1) achieves the worst performance, which means the simpler sample should be trained in the beginning stage or the middle stage.

**Evolution of Accuracy Along with the Training Steps.** We show the accuracy evolution of LLaMA on different benchmarks during training in Figure 2. For all four datasets, our method steadily outperforms the baseline when training converges, indicating better learning performance. In addition, except for RobustLR, we achieve consistently better results at the early stage of training compared to the baselines, demonstrating the effectiveness of our approach.

**Sensitivity Analysis.** To investigate which complexity (symbolic logical form or LLM logical reasoning) is more essential in the sample splitting, we conduct the sensitivity analysis on the weight parameter $\alpha$, as shown in Figure 3. We find that a small $\alpha$ is the best choice for Ministral, and a large $\alpha$ is the best choice for LLaMA, showing there is no general consistent choice of $\alpha$ for different datasets and backbones, and indicating that both complexity are important.

## 6 RELATED WORK

**Logical Reasoning in LLMs.** Methods for enhancing the logical reasoning capabilities of large language models can be broadly categorized into three types. Solver-based approaches first prompt LLMs to convert natural language problems into symbolic expressions, then leverage a corresponding logic solver to infer the answer (Ye et al., 2023; Pan et al., 2023; Olausson et al., 2023). Prompt-based methods follow two main strategies. The first explicitly generates logical chains during question

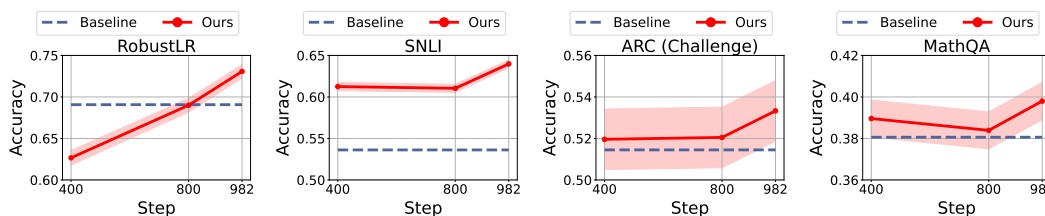

Figure 2: The evolution of accuracy of LLaMA on logical and other reasoning benchmarks along with the training steps. The dashed blue lines represent the performance of baseline.

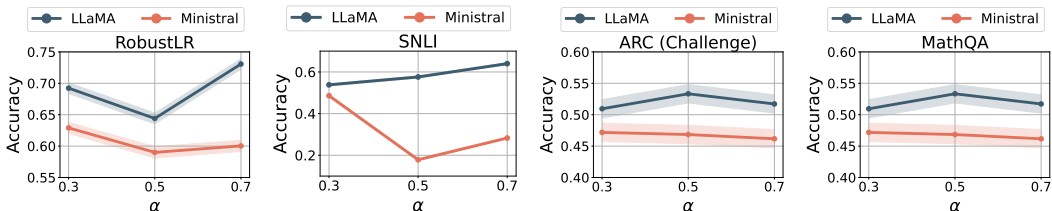

Figure 3: Sensitivity analysis on the weight parameter $\alpha$. The blue line represents the results of LLaMA, while the orange line represents the results of Ministral.

answering (Wei et al., 2022; Yao et al., 2024; Zhang et al., 2024). The second strategy leverages LLMs through prompts to symbolize natural language, reason step by step, and verify results (Xu et al., 2024; Liu et al., 2024; Li et al., 2024b). Fine-tuning approaches construct synthetic datasets that contain clear logical reasoning processes (Bao et al., 2024; Morishita et al., 2024), or argument the training dataset with logical reasoning data (Feng et al., 2024; Wan et al., 2024; Jiao et al., 2024) to fine-tune the LLMs to improve the accuracy of logic reasoning. While previous approaches that fine-tune LLMs in a single stage with logical reasoning datasets, our work introduces a phased fine-tuning method to train LLMs progressively from simple to complex logical reasoning samples.

**Curriculum Learning.** Curriculum Learning (CL) (Bengio et al., 2009), inspired by the human learning process, is a training strategy in machine learning that presents data in a "from easy to hard" manner, significantly improving both generalization and training efficiency (Campos, 2021; Wang et al., 2020). Pang et al. (2024) and Huang & Xiong (2024) both design instruction fine-tuning strategies that organize data from easy to hard by models' feedback, demonstrating gains across multiple tasks of LLMs. TAPIR (Yue et al., 2024) also uses a strong oracle LLM to iteratively select and refine increasingly difficult samples, systematically escalating task difficulty during training. Yang et al. (2024b) study fine-tuning based on CL in medical QA and observe modest accuracy gains from sorting examples by difficulty. Wu et al. (2024) and Varshney et al. (2022) design a CL training framework based on data quality or model uncertainty, boosting performance respectively on visual instruction and multiple NLP tasks. Our paper first introduces phased fine-tuning based on curriculum learning into logical QA tasks to significantly improve the logical reasoning ability of LLMs.

## 7 CONCLUSION

This study focuses on logic QA tasks and explores employing phased instruction fine-tuning to enhance the logical reasoning ability of large language models. We introduced a method for computing sample difficulty by combining the symbolic logical form complexity and the LLM logical reasoning complexity of logic QA tasks. This enables automatic difficulty measurement directly from propositional logic data, without relying on external difficulty annotations. We carefully constructed a training set of propositional logic QA samples based on the public FLDx2 dataset and conducted evaluations on multiple benchmark datasets, covering logical task RobustLR, LogicNLI, SNLI, RTE, FOLIO, and LogiQA2.0, and general reasoning tasks MMLU, Big-bench, ARC, and MathQA. Experimental results demonstrate the effectiveness of our phased fine-tuning approach for logical reasoning and general reasoning, as well as the reliability of our proposed automatic difficulty measurement method. One possible limitation of this paper is that the current phased fine-tuning approach focuses only on basic propositional logic, and can be further extended to more complex logic systems in the future, including first-order logic, modal logic, and higher-order logic.

## REPRODUCIBILITY STATEMENT

The dataset preprocessing procedures, experimental details and evaluation metrics are described in the main text. Complete training code are provided in an anonymized repository at `https://anonymous.4open.science/r/ICLR26_23017-FF14`.

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

## USAGE OF LLMS

In this work, we limitedly use LLMs as an assistive writing tool. Specifically, we used LLMs to replace words with synonyms, restructure sentences, and check grammars in the paragraphs. All significant contributions, research ideas, experiments, analyses, and final writing decisions were made by the authors. The authors understand that they take full responsibility for the contents written in this paper.

## A   LOGICAL CONNECTIVES

Table 4: Definition of logical connectives.

| Logical Connective | Name | Meaning in Natural Language |
|---|---|---|
| $\neg$ | Negation | It is not the case that ... |
| $\wedge$ | Conjunction | Both ... and ... |
| $\vee$ | Disjunction | Either ... or ... |
| $\rightarrow$ | Implication | If ... then ... |
| $\leftrightarrow$ | Biconditional | ... if and only if ... |

Table 5: Truth table of logical connectives.

| $P$ | $Q$ | $\neg P$ | $P \wedge Q$ | $P \vee Q$ | $P \rightarrow Q$ | $P \leftrightarrow Q$ |
|---|---|---|---|---|---|---|
| T | T | F | T | T | T | T |
| T | F | F | F | T | F | F |
| F | T | T | F | T | T | F |
| F | F | T | F | F | T | T |

## B   PROMPTS FOR TRANSLATING NATURAL LANGUAGE TO SYMBOLIC LANGUAGE

## C   STATISTICAL DESCRIPTION OF DATASETS

We observe the formal complexity $\mathcal{C}_{SL}$ and other potential difficulty indicators of the samples, including the number of premises, types of atomic propositions, and the DNF length we defined. Their frequency distributions are shown in Figure 5. It can be seen that our defined formal complexity $\mathcal{C}_{SL}$ provides good discrimination and has a distribution closer to normal compared to the other two indicators.

The resulting frequency distribution of sample complexity in propositional logic QA is shown in Figure 6a, and the cumulative distribution function estimated via kernel density estimation is shown in Figure 6b. It can be seen that the final sample complexity is approximately normally distributed, with a maximum around 0.7, indicating that the rankings of formal and natural language complexities are not entirely consistent.

## D   BENCHMARK DATASETS

- FOLIO dataset (Han et al., 2024) evaluates the first-order logical reasoning abilities.
- SNLI dataset (Bowman et al., 2015) evaluates the abilities of recognizing entailment relations between short text pairs.
- MMLU dataset (Hendrycks et al., 2021) evaluates knowledge and reasoning abilities across 57 diverse tasks at high school, college, and professional levels.

**Step1: Extracting Atomic Propositions from the Premises**

• **Task**
- Destract atomic propositions from the following facts and assign symbols;

• **Input**
- Fact1: If Tom read Game Theory, then he gained knowledge.
- Fact2: If Tom gains knowledge, he becomes smarter.
- Fact3: Tom read Game Theory.

• **Output**
- {"symbol1": "proposition1", "symbol2": "proposition2", ...} (use P, Q, R as symbols, and ¬ for negation)

**Step2: Translating Premises and Conclusion**

• **Task**
- Generate propositional logic formula based on the mapping of atomic propositions

• **Mapping**
{
  "P": "Tom reads "Game Theory",
  "Q": "Tom gains knowledge",
  "R": "Tom becomes smarter",
}

• **Sentences**
- Fact1: If Tom read Game Theory, then he gained knowledge.
- Fact2: If Tom gains knowledge, he becomes smarter.
- Fact3: Tom read Game Theory.

• **Output**
- only output the symbolic formula such as $(P \land Q) \rightarrow R$

LLM

LLM

"P": "Tom reads "Game Theory"
"Q": "Tom gains knowledge"
"R": "Tom becomes smarter"

$$P \rightarrow Q$$
$$Q \rightarrow R$$
$$P$$

Figure 4: Example of prompts and outputs for **step1:** extracting atomic propositions from the premises and **step2:** translating premises and conclusion based on the mapping of atomic propositions.

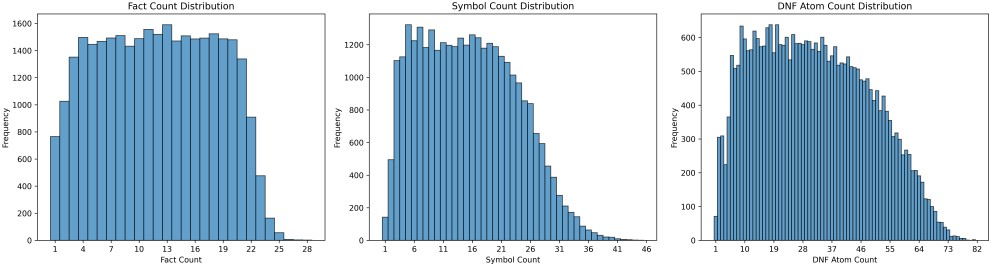

Figure 5: Histogram of the frequency distribution of the number of premises, propositional variables, and atomic propositions in DNF

- Big-bench dataset (Srivastava et al.) comprises 204 diverse tasks spanning linguistics, math, common-sense reasoning. We perform CoT evaluation in zeroshot setting with Big-bench.

- RobustLR dataset (Sanyal et al., 2022) tests abstract and compositional logical reasoning by multiple-choice questions. We construct the data into multiple choice questions with RobustLR.

- LogicNLI dataset (Tian et al., 2021) is an NLI-style benchmark designed to diagnose language models' first-order logic reasoning ability. We construct the data into multiple choice questions with LogicNLI.

- RTE dataset (Dagan et al., 2005; Giampiccolo et al., 2007; Bentivogli et al., 2009) comes from a series of textual entailment reasoning tasks. We construct the data into multiple choice questions with RTE.

- ARC (Challenge) (Clark et al., 2018) is a more challenging subset in a benchmark of grade-school science questions designed to test knowledge-intensive and reasoning-heavy question answering.

- MathQA dataset (Amini et al., 2019) is a large-scale collection of math word problems requiring interpretable and accurate neural reasoning.

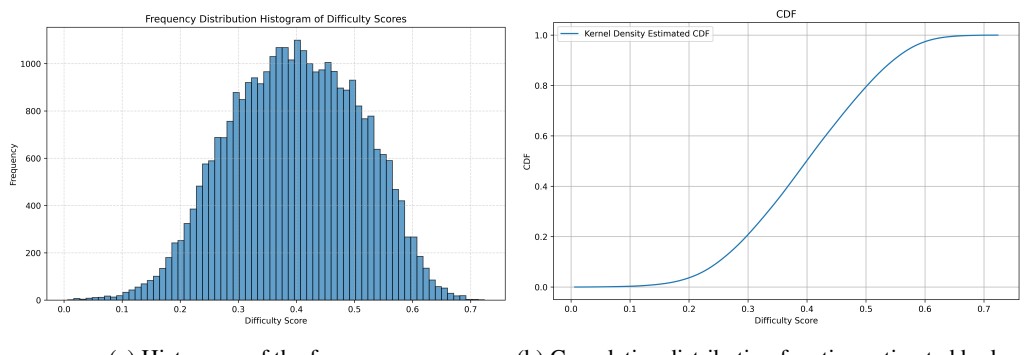

(a) Histogram of the frequency

(b) Cumulative distribution function estimated by kernel density

Figure 6: Statistical description of logic QA sample complexity.

- LogiQA2.0 dataset (Liu et al., 2023) evaluates complex logical reasoning abilities with 35k premise–hypothesis pairs.

## E  PROOF SKETCH

**Proposition 3.1.** $\succ$ defines a partial order on the set $S$.

*Proof.* Consider any $(m_1, m_2, \ldots, m_n) \in S$. Clearly, we have $n \geq n$, $m_1 \geq m_1$, $m_2 \geq m_2$, $\ldots$, $m_n \geq m_n$, hence $q \succ q$, and reflexivity holds.

Consider any $(m_1, m_2, \ldots, m_n)$, $(m'_1, m'_2, \ldots, m'_{n'}) \in S$. If $(m_1, m_2, \ldots, m_n) \succ (m'_1, m'_2, \ldots, m'_{n'})$ and $(m'_1, m'_2, \ldots, m'_{n'}) \succ (m_1, m_2, \ldots, m_n)$ both hold, then $n \geq n'$ and $n' \geq n$, so $n = n'$; similarly, from $m_1 \geq m'_1$, $\ldots$, $m_{n'} \geq m'_{n'}$ and $m'_1 \geq m_1$, $\ldots$, $m'_n \geq m_n$, we have $m_1 = m'_1$, $\ldots$, $m_{n'} = m'_{n'}$, thus $(m_1, m_2, \ldots, m_n) = (m'_1, m'_2, \ldots, m'_{n'})$, and antisymmetry holds.

Consider any $(m_1, m_2, \ldots, m_n)$, $(m'_1, m'_2, \ldots, m'_{n'})$, $(m''_1, m''_2, \ldots, m''_{n''}) \in S$. If $(m_1, m_2, \ldots, m_n) \succ (m'_1, m'_2, \ldots, m'_{n'})$ and $(m'_1, m'_2, \ldots, m'_{n'}) \succ (m''_1, m''_2, \ldots, m''_{n''})$, then $n \geq n'$ and $n' \geq n''$, hence $n \geq n''$; similarly, from $m_1 \geq m'_1$, $\ldots$, $m_{n'} \geq m'_{n'}$ and $m''_1 \geq m''_1$, $\ldots$, $m''_{n''} \geq m''_{n''}$, we get $m_1 \geq m''_1$, $\ldots$, $m_{n''} \geq m''_{n''}$, i.e., $(m_1, m_2, \ldots, m_n) \succ (m''_1, m''_2, \ldots, m''_{n''})$, and transitivity holds.

In conclusion, $\succ$ defines a partial order on $S$. $\qquad\square$

**Theorem 4.1.** At step $t$, the expected convergence rate for training sample $S_t$ is monotonically decreasing with the Difficulty Score $\Psi(S_t)$. Formally, we have

$$\frac{\partial \Delta_t(\Psi)}{\partial \Psi} = -8\eta^2 \mathbb{E}_{S_t \sim \mathcal{D}|\Psi} \left[ \|S_t\|^2 \right] \Psi \leq 0.$$

*Proof.* Let the last-layer representation of LLM be $x_i$ for training sample $i$. We require another assumption that the probability of label $Y$ depends only on the loss $L(S_t, \bar{\mathbf{w}})$ and is independent of $x$ to prove this theorem.

We formulate the logic QA task as a regression problem that predicts the probability of the conclusion being true. Then the loss can be written as:

$$L(S_t, \mathbf{w}) = (x \cdot \mathbf{w} - y)^2.$$

Let $s$ denote the gradient vector at step $t$, then the updating equation for the parameter vector is:

$$\mathbf{w}_{t+1} = \mathbf{w}_t - 2\eta(x \cdot \mathbf{w} - y)x = \mathbf{w}_t + s \tag{4}$$

$$s \doteq -2\eta(x \cdot \mathbf{w} - y)x \tag{5}$$

Let $\Omega_x$ denote the hyperplane on which the gradient $s$ vanishes, i.e., $s = 0$. Let $\bar{z}$ be the projection of $\bar{w}$ on $\Omega_x$. The Difficulty Score of $S_t$ is $\Psi(S_t) = L(S_t, \bar{w}) = L(S_t, \bar{z} + (\bar{w} - \bar{z})) = ||S_t||^2 ||\bar{w} - \bar{z}||^2$.

Next, we embed the data points in the parameters space, representing each datapoint $\mathbf{x}$ using a hyperspherical coordinate system $[r, \vartheta, \Phi]$, with pole (origin) fixed at $\bar{w}$ and polar axis (zenith direction) $\vec{\mathcal{O}} = \bar{w} - w_t$. $r$ denotes the vector's length, while $0 \leq \vartheta \leq \pi$ denotes the polar angle with respect to $\vec{\mathcal{O}}$. Let $\Phi = [\varphi_1, \ldots, \varphi_{d-1}]$ denote the remaining polar angles.

We introduce the notation $\lambda = ||\bar{w} - w_t||$. Let $s_{\mathcal{O}}$ denote the projection of the gradient vector $s$ on the polar axis $\vec{\mathcal{O}}$, and let $s_{\perp}$ denote the perpendicular component. From the updating equation and the definition of $\Psi$, we have $s = -2\eta x(x \cdot w_t - y) = -2\eta x(x \cdot (w_t - \bar{w}) \pm \Psi)$, and

$$s_{\mathcal{O}} = s \cdot \frac{\bar{w} - w_t}{\lambda}$$
$$= 2\frac{\eta}{\lambda} \left[ r^2 \lambda^2 \cos^2 \vartheta \mp \Psi r \lambda \cos \vartheta. \right]$$

Then, the convergence rate at $w_t$ given $\Psi$ is:

$$\Delta(\Psi) = (-\lambda)^2 - \mathbb{E}_{S_t \sim \mathcal{D}|\Psi}[(-\lambda + s_{\mathcal{O}})^2 + s_{\perp}^2]$$
$$= \lambda^2 - \left( \lambda^2 - 2\lambda \mathbb{E}_{S_t \sim \mathcal{D}|\Psi}[s_{\mathcal{O}}] + \mathbb{E}_{S_t \sim \mathcal{D}|\Psi}[s_{\mathcal{O}}^2] \right) - \mathbb{E}_{S_t \sim \mathcal{D}|\Psi}[s_{\perp}^2]$$
$$= 2\lambda \mathbb{E}_{S_t \sim \mathcal{D}|\Psi}[s_{\mathcal{O}}] - \mathbb{E}_{S_t \sim \mathcal{D}|\Psi}[s^2]$$

To simplify the notations, henceforth $\mathbb{E}$ stands for $\mathbb{E}_{S_t \sim \mathcal{D}|\Psi}$. In addition, we define a shorthand notation $(\pm \Psi)$ to be used inside the expectation operator $\mathbb{E}[\cdot]$. It conveys that the operand of $\mathbb{E}[]$ should be multiplied by either $+\Psi$ or $-\Psi$, depending on whether the label $y$ equals $x \cdot \bar{w} + \Psi$ or $x \cdot \bar{w} - \Psi$ respectively. When expectation is computed, each case is assigned the conditional probability of the corresponding label as defined above. Then we have:

$$\Delta(\Psi) = 4\eta \mathbb{E}[r^2 \lambda^2 \cos^2 \vartheta] - 4\eta^2 \mathbb{E}[r^4 \lambda^2 \cos^2 \vartheta] - 4\eta^2 \Psi^2 \mathbb{E}[r^2] - 4\eta \mathbb{E}[(\pm \Psi) r \lambda \cos \vartheta]$$
$$- 8\eta^2 \mathbb{E}[(\pm \Psi) r^3 \lambda \cos \vartheta].$$

By the additional assumption about the distribution of $Y$, we have

$$\mathbb{E}[(\pm \Psi) r \lambda \cos \vartheta] = \mathbb{E}[(\pm \Psi) r^3 \lambda \cos \vartheta] = 0,$$

from which we can infer that

$$\Delta(\Psi) = 4\eta \mathbb{E}[r^2 \lambda^2 \cos^2 \vartheta] - 4\eta^2 \mathbb{E}[r^4 \lambda^2 \cos^2 \vartheta] - 4\eta^2 \Psi^2 \mathbb{E}[r^2].$$

As a consequence, we have

$$\frac{\partial \Delta_t(\Psi)}{\partial \Psi} = -8\eta^2 \mathbb{E}_{S_t \sim \mathcal{D}|\Psi} \left[ ||S_t||^2 \right] \Psi \leq 0,$$

which concludes the proof of the theorem. $\square$

## F   COMPARISON BETWEEN CNF AND DNF

**Definition F.1.** *A conjunctive normal form (CNF) is a conjunction of disjunctions, where each disjunction consists of atomic propositions or their negations. It is formally expressed as:*

$$q = \bigwedge_{i=1}^{n} \left( \bigvee_{j=1}^{m_i} e_{ij} \right),$$

*where $e_{ij}$ is an atomic proposition or the negation of an atomic proposition.*

In practice, transforming formula 1 into CNF incurs significantly higher time and space complexity than DNF, and the extreme length expansion makes CNF unsuitable for defining syntactic complexity.

As shown in Table 6, for the first nine samples in our dataset (the tenth sample contains 22 premises and could not be converted to CNF within a reasonable time), the average length of CNF is over 500 times that of DNF, and the average CPU runtime is more than three times longer. Compared to the number of original premises, CNF length expands by a factor of 500, whereas DNF length increases only by a factor of 2. Therefore, we adopt DNF as the canonical form.

Table 6: Statistics on the time and space complexity of converting to DNF and CNF

| Canonical Form | Avg. Length | Avg. Runtime | Avg. #Premises | Max #Premises | Avg. Length / #Premises |
|---|---|---|---|---|---|
| DNF | 19.2 | 1.12e-6s | 8.2 | 18 | 2.53 |
| CNF | 9568.5 | 5.48e-7s | 8.2 | 18 | 577.85 |

## G  COST AND RUNTIME ANALYSIS

Our proposed method involves some extra computation via API or on a local GPU server. The token consumption in the step translating natural language to symbolic language is 25,091,708 in total. Note that the translation should only be performed once; then, the results can be stored for the complexity labeling and training. The typical runtime by step can be seen in Table 7.

Table 7: Runtime cost of the whole process including translating natural language to symbolic language, complexity labeling, and fine-tuning training.

| Step | Translation | $\mathcal{C}_{SL}$ | $\mathcal{C}_{NL}$ | Training |
|---|---|---|---|---|
| Time | 8.2 h | 2 min | 23.5 min | 2 h 23 min |

## H  EXTRA EXPERIMENTAL RESULTS

We conduct more experiments on the Ministral model. The first one is the phased fine-tuning for all permutations of training order, which is shown in Table 8. The second one is about an additional performance comparison, with a naive baseline using the prompt-based method to obtain the difficulty of all samples and perform phased fine-tuning. The result can be seen in Table 9. Our method significantly outperforms this baseline on almost all benchmarks except for MathQA.

Table 8: Performance of Ministral under different instruction training orders. Background cell colors range from light to dark, indicating increasing values within each row.

| Benchmark | Metric | 1-2-3 | 2-1-3 | 3-1-2 | 1-3-2 | 2-3-1 | 3-2-1 |
|---|---|---|---|---|---|---|---|
| MMLU | acc | **0.506 ± 0.004** | 0.429 ± 0.004 | 0.458 ± 0.004 | 0.469 ± 0.004 | 0.478 ± 0.004 | 0.487 ± 0.004 |
| Big-bench | exact_match | **0.116 ± 0.003** | 0.074 ± 0.003 | 0.081 ± 0.003 | 0.073 ± 0.003 | 0.106 ± 0.004 | 0.092 ± 0.003 |
| RobustLR | acc | 0.629 ± 0.008 | 0.457 ± 0.009 | 0.621 ± 0.009 | **0.689 ± 0.008** | 0.428 ± 0.009 | 0.509 ± 0.009 |
| LogicNLI | acc | **0.440 ± 0.011** | 0.364 ± 0.011 | 0.292 ± 0.010 | 0.386 ± 0.011 | 0.350 ± 0.010 | 0.284 ± 0.010 |
| SNLI | acc | **0.485 ± 0.005** | 0.336 ± 0.004 | 0.373 ± 0.004 | 0.345 ± 0.004 | 0.197 ± 0.004 | 0.435 ± 0.005 |
| RTE | acc | 0.820 ± 0.007 | 0.762 ± 0.008 | **0.823 ± 0.007** | 0.594 ± 0.009 | 0.821 ± 0.007 | 0.808 ± 0.007 |
| ARC (challenge) | acc | 0.471 ± 0.014 | 0.446 ± 0.014 | 0.473 ± 0.014 | 0.466 ± 0.014 | **0.474 ± 0.014** | 0.470 ± 0.014 |
| MathQA | acc_norm | 0.341 ± 0.008 | 0.339 ± 0.008 | 0.343 ± 0.008 | **0.350 ± 0.008** | 0.339 ± 0.008 | 0.345 ± 0.008 |
| FOLIO | exact_match | 0.503 ± 0.015 | 0.407 ± 0.015 | 0.483 ± 0.015 | 0.493 ± 0.015 | 0.472 ± 0.015 | **0.512 ± 0.015** |
| LogiQA2.0 | acc_norm | **0.306 ± 0.011** | 0.306 ± 0.011 | 0.297 ± 0.011 | 0.306 ± 0.011 | 0.302 ± 0.011 | 0.288 ± 0.011 |

## I  RELATED WORK

**Curriculum learning.** Curriculum Learning (CL), inspired by the human learning process (Bengio et al., 2009), is a training strategy in machine learning that presents data in a "from easy to hard" manner, significantly improving both generalization and training efficiency. In computer vision,

Table 9: Comparison of our phased instruction fine-tuning and phased fine-tuning with prompt-based difficulty labeling baselines across different reasoning benchmark datasets. RI denotes *relative improvement* over the corresponding baseline.

| Benchmark | Metric | Ministral-P | Ministral-PFT | RI |
|---|---|---|---|---|
| MMLU | acc | $0.435 \pm 0.004$ | $\mathbf{0.506 \pm 0.004}$ | 16.322% ↑ |
| BIG-bench | exact_match | $0.089 \pm 0.003$ | $\mathbf{0.116 \pm 0.003}$ | 30.337% ↑ |
| RobustLR | acc | $0.581 \pm 0.009$ | $\mathbf{0.629 \pm 0.008}$ | 8.262% ↑ |
| LogicNLI | acc | $0.367 \pm 0.011$ | $\mathbf{0.440 \pm 0.011}$ | 19.891% ↑ |
| SNLI | acc | $0.218 \pm 0.004$ | $\mathbf{0.485 \pm 0.005}$ | 122.477% ↑ |
| RTE | acc | $0.804 \pm 0.008$ | $\mathbf{0.820 \pm 0.007}$ | 1.990% ↑ |
| ARC (challenge) | acc | $0.453 \pm 0.014$ | $\mathbf{0.471 \pm 0.014}$ | 3.974% ↑ |
| MathQA | acc_norm | $\mathbf{0.348 \pm 0.008}$ | $0.341 \pm 0.008$ | 2.011% ↓ |
| FOLIO | exact_match | $0.504 \pm 0.015$ | $0.492 \pm 0.015$ | 2.236% ↑ |
| LogiQA2.0 | acc_norm | $0.298 \pm 0.011$ | $\mathbf{0.306 \pm 0.011}$ | 2.685% ↑ |

CL has been successfully applied to tasks such as image classification and object detection. For example, it improves model performance by gradually increasing image complexity (Soviany et al., 2020), mitigates the impact of noisy data in weakly supervised settings by filtering high-confidence samples (Jiang et al., 2018), and enhances image quality in GANs through progressive generation strategies (Karras et al., 2017). Natural language processing (NLP) is one of the most successful domains for curriculum application. In machine translation, curriculum designs based on sentence length or complexity reduce training time by 70% and improve BLEU scores by 2.2 points (Platanios et al., 2019). CL also significantly improves model performance in reading comprehension (Tay et al., 2019), relation extraction (Huang & Jia, 2019), and other tasks by dynamically adjusting data difficulty.

CL has also demonstrated unique advantages in multimodal learning (Gong et al., 2019) and reinforcement learning (Florensa et al., 2017). However, its broader adoption still faces major challenges. On the one hand, curriculum design heavily relies on domain knowledge, while automatic curriculum generation methods (e.g., self-paced learning and RL-based teacher models) often suffer from high computational cost and sensitivity to hyperparameters. On the other hand, curriculum strategies are highly task-dependent, lacking unified theoretical guidance, and existing research lacks clear definitions of "easy" and "hard" samples, as well as standardized benchmarks for evaluating curriculum effectiveness (Wang et al., 2020).

Despite its strong empirical success in machine learning, current mainstream instruction finetuning approaches, such as LIMA, Alpaca, Alpagasus, and Superfiltering (Li et al., 2023; 2024a), typically adopt a one-off instruction fine-tuning paradigm on entire instruction datasets, which are characterized by high quality and diversity. These methods focus on proposing various techniques for data selection, construction, and filtering to optimize datasets for better fine-tuning performance with high-quality, high-difficulty, and diverse data (Wang et al., 2023). However, they overlook the internal complexity of the instruction set itself. As a result, the one-off fine-tuning paradigm fails to sufficiently equip models with the ability to precisely understand and execute diverse instructions. Although curriculum learning has been applied in some LLMs training mehods, as mentioned in Section 6 of the main text, it has not yet been considered for enhancing logical reasoning capabilities in LLMs.

In this paper, we propose a novel difficulty measurement of logical QA and then we finetune the pre-trained models progressively on the ordered sequence focusing on the LLM's logical reasoning abilities.

**Logical reasoning in LLMs.** Methods for enhancing the logical reasoning capabilities of large language models can be broadly categorized into three types: approaches based on external solvers, prompt design, and fine-tuning. The solver-based approaches usually prompt the model to convert natural language problems into symbolic expressions, then call a corresponding logic solver for logical reasoning, and finally generate the answer using ensemble methods such as majority voting. Main methods following this approach include Satlm (Ye et al., 2023), LOGIC-LM (Pan et al., 2023), LINC (Olausson et al., 2023), etc.

Prompt-based methods follow two main strategies. The first explicitly models logical chains during question answering, such as CoT (Wei et al., 2022), ToT (Yao et al., 2024), DoT (Zhang et al., 2024). The second strategy prompts the LLMs sequentially perform tasks such as symbolizing natural language problems, decomposing the task, reasoning step by step, and verifying results. Main methods based on this approach include SymbCoT (Xu et al., 2024), Logic-of-Thought (Liu et al., 2024), LINA (Li et al., 2024b), etc.

The limitations in the reasoning abilities of large language models can largely be attributed to the lack of high-quality reasoning samples (especially multi-step logical deductions or proofs) in the pretraining corpora (Morishita et al., 2024). Fine-tuning approaches address this by constructing synthetic datasets that contain clearly presented logical reasoning processes, or by collecting large numbers of logical reasoning steps to augment the training data, thereby fine-tuning the models to improve both accuracy and interpretability.

A logic-driven contrastive learning approach (Wang et al., 2022) and a data augmentation method, AMR-LDA (Bao et al., 2024), aim to augment logical reasoning datasets by leveraging structured semantic representations and logic-modified AMR graphs. To increase interpretability, LOGIPT (Feng et al., 2024) simulates the reasoning process of the Pyke solver and is fine-tuned on an instruction dataset aligning natural language problems with symbolic reasoning steps. Similarly, ALT (Morishita et al., 2024) creates a synthetic logic corpus from deduction steps, using them to fine-tune models for step-wise reasoning. LogicAsker (Wan et al., 2024) builds a skill set grounded in formal logic, generates corresponding natural language tasks, and adaptively fine-tunes LLMs by diagnosing weaknesses in reasoning abilities. To minimize annotation cost, LogicLLM (Jiao et al., 2024) proposes a fully self-supervised framework for integrating logical reasoning into LLMs.

However, solver-based methods are vulnerable to errors in translation and face challenges such as search space explosion when handling complex logical reasoning problems. Prompt-based methods depend on the model's initial reasoning capabilities from pretraining, but empirical evidence suggests that models like LLaMA-13B achieve only slightly better than random performance on logical question-answering tasks when relying solely on in-context learning. Moreover, we observe that directly fine-tuning large models often fails to effectively address complex logical reasoning tasks, highlighting the need to develop more efficient fine-tuning paradigms tailored specifically for enhancing logical reasoning in large language models. This paper addresses the limitations of existing one-off fine-tuning approaches by applying a phased fine-tuning methods inspired by curriculum learning. We carefully design a difficulty measurement of logical QA and a phased partitioning strategy, aiming to improve fine-tuning efficiency through phased training and effectively enhance the logical reasoning ability of large language models.

## J  BROADER IMPACTS

Our work aims to enhance the logical reasoning abilities of large language models through a phased fine-tuning framework guided by propositional logic difficulty. On the positive side, this advancement may increase the reliability and trustworthiness of LLMs in critical applications such as education, legal document analysis, and scientific research, where rigorous logical reasoning is essential. It could also benefit downstream systems in fields requiring interpretable and structured reasoning, supporting more transparent decision-making processes.

However, we also recognize several potential risks. Improving logical inference capabilities could inadvertently strengthen the use of LLMs in generating misleading arguments or disinformation that appear logically coherent, which may increase the difficulty of detection. Additionally, in domains where fairness or ethical norms intersect with logic (e.g., automated legal judgments), stronger logical reasoning might be used to justify biased conclusions if the model is trained on skewed or flawed data.

To mitigate these risks, we recommend pairing our approach with robust data auditing, bias detection mechanisms, and transparency practices that ensure the models' reasoning chains can be inspected. Furthermore, the release of models trained using our methodology should consider controlled environments or usage agreements to discourage misuse.

