# OpenReview forum: "Enhancing Logical Reasoning of Large Language Models via Phased Fine-Tuning"
_ICLR.cc/2026/Conference — Submitted to ICLR 2026_

### Official Review · Reviewer_MGEu · 2025-10-30

**Soundness:** 3
**Presentation:** 3
**Contribution:** 3
**Rating:** 4
**Confidence:** 5

**Summary:**

This paper focuses on the logical reasoning capabilities of LLMs, by using curriculum learning method, a three-stage training framework from easy to hard was established, and it was proposed to use the combination of syntactic complexity and semantic understanding difficulty to distinguish the difficulty, achieving the division of three stages. Additionally, the authors trained models based on Llama-3.1 8B and Ministral-8B on a training set produced by the pipeline, and results show that the framework enhances logic reasoning abilities.

**Strengths:**

1. The combination of symbolic (DNF-based) and probabilistic (entropy-based) reasoning complexity is theoretically well-motivated and interpretable.

2. The phased fine-tuning design aligns with human-style curriculum learning and shows stable performance gains across datasets.

3. The method is evaluated on diverse benchmarks, showing both logical and general reasoning improvements.

**Weaknesses:**

1. The paper only shows that different models prefer different $\alpha$ values but does not analyze the underlying reasons or training dynamics, making this section observational rather than explanatory.

2. The claimed “improved fine-tuning efficiency” is not supported by any efficiency analysis or quantitative evidence.

3. This method is primarily tested on formal logic QA datasets. Its applicability to more naturalistic or open-domain reasoning tasks remains uncertain.

4. No phase-wise results are provided to verify that each training stage contributes progressively.

5. Minor formatting issue (line 258, Appendix B).

**Questions:**

1. How reliable is the LLM-based natural-to-symbolic translation process?

2. How does the framework handle ambiguous or underspecified sentences during translation?

3. How well would the method generalize to more naturalistic reasoning datasets?

4. Line 420, in “sensitive analysis”, can you obtain more accurate conclusions by further analyzing through trying more model sizes and series?

5. Line 296, the weight parameter  $\alpha$  do not have specific values. Considering that “sensitivity analysis” were conducted later, weight parameters  $\alpha $  have a significant impact on the final performance of the model. Therefore, it is necessary to specify what the weight parameters were for the experimental settings in other experiments.

6. Figure 2 shows that the model accuracy will improve as the number of training steps increases, but it is clear that there is a significant turning point at 800 steps, which requires further analysis and explanation. Also, why was it chosen to stop at 982 steps? What will happen to the accuracy if training continues further?

---

### Official Review · Reviewer_ewk3 · 2025-10-30

**Soundness:** 3
**Presentation:** 3
**Contribution:** 3
**Rating:** 6
**Confidence:** 4

**Summary:**

This paper addresses the limited logical reasoning capabilities of large language models (LLMs) by proposing a phased fine-tuning framework based on curriculum learning. The authors focus on propositional logic question-answering (QA) tasks and introduce a novel method for automatically measuring sample difficulty using two components.

**Strengths:**

1) The paper introduces a principled approach to quantifying the difficulty of logical reasoning samples, combining both syntactic (DNF-based) and semantic (LLM-prediction-based) complexities. This is a clear advancement over heuristic or manual difficulty labeling, and the theoretical grounding in propositional logic and information theory is solid.
2) The proposed method is extensively evaluated across a wide range of benchmarks, including both logical reasoning (e.g., FOLIO, LogicNLI) and general reasoning (e.g., MMLU, ARC) tasks.
3) The paper provides theoretical justification for the proposed method.

**Weaknesses:**

1) The entire pipeline is tightly coupled to propositional logic: DNF conversion, independence of atomic propositions, and entropy computation all break down once quantifiers, equality, or modal operators appear. The paper title promises “logical reasoning” in general, yet the method cannot handle even minimal first-order problems.
2) The approach relies on an LLM to translate natural language premises and conclusions into symbolic form. Errors in this translation step could propagate into the difficulty measurement and fine-tuning process. The paper does not thoroughly analyze the impact of translation errors or propose robust solutions for handling them.
3) The weight $(\alpha\)$ balancing $\(C_{SL}\)$ and $\(C_{NL}\)$  is shown to be dataset- and model-dependent, with no consistent optimal value. This suggests that the method may require careful tuning for new tasks or models, limiting its out-of-the-box applicability.

**Questions:**

1) The paper explicitly limits itself to propositional logic. How would you extend the difficulty metric to first-order or modal logic without incurring exponential blow-ups during normal-form conversion?
2) You provide a strong justification for using DNF over CNF due to computational explosion. However, the "difficulty" of a logical problem might also be influenced by its inherent structure, which could be more naturally captured by other forms (e.g., the number of nested implications). Did you explore any other canonical forms or syntactic complexity measures beyond DNF length?

---

### Official Review · Reviewer_Uno8 · 2025-11-01

**Soundness:** 2
**Presentation:** 3
**Contribution:** 3
**Rating:** 2
**Confidence:** 4

**Summary:**

This work proposes a difficulty measurement scheme for propositional-logic-structured problems in order to order training samples from easy to hard during fine-tuning. The method is based on a semantic complexity measure over the LLMs independent probability evaluations of each premise in the problem, and a structural complexity measure taken as the length of the problem in DNF form.

**Strengths:**

1. The paper is well laid out and mostly easy to follow
2. Conceptually, the rationale behind the method, and the method itself, is mostly clear
3. The method is benchmarked over a good variety of datasets
4. Sensitivity analyses are provided for the hyper-parameters of the method

**Weaknesses:**

1. Some minor claims made in the paper are unsupported (Questions 3, 9, 13)
2. Many of the formalisms presented in the methodology section are poorly defined or mostly extraneous to the method (Questions 3,4,5,6,7,8)
3. Some elements of experimental design seem questionable:
     a. Baseline selection is sparse, only comparing to the most naive approach (vanilla fine-tuning) (Question 10)
     b. metric choice on some benchmarks is not self-explanatory, and no explanation is provided (Question 11)
4. Implementation details are not provided: code is not provided, and the details of how the LLM's individual proposition probabilities are extracted are not provided (contextualization, prompting, post-model normalization over the support, ...)

**Questions:**

1. l. 269: can you define the w described as a condition to the independence?
2. l. 269: to my knowledge, independence is not a natural concept in propositional logic. Can you define independence here?
3. l. 303: "indicating that the rankings [...] are not entirely consistent". Can you explain how this is indicated? I see no indication of this. We would need to see some kind of (empirical) problem-wise score difference distribution to validate this claim.
4. Section 3.2: you define the Difficulty Partial Order Principle (DPOP), arguing that problems which do not satisfy it are incomparable (l. 235). However, since you prove only that your later complexity measurement function is such that all rankings from the direct DPOP are preserved, your measurement function will also capture rankings which come from problem pairs which do not satisfy DPOP. This is clear from the fact that a problem with a single clause of 1000 variables will be seen as more complex than a problem with 5 clauses of 5,4,3,2,1 lengths respectively, whereas these two problems do not satisfy DPOP. Since later on you do not refer back to DPOP, what was the purpose of its introduction? It seems likely that most of the pairwise difficulty orderings determined by your measurement function will in practice not observe DPOP, since it is a fairly strict rule.
5. I find your probabilistic framing of propositional logic in the logic QA setting somewhat confusing. If the context to a problem gives "Tom reads Game Theory" concretely, the probability p(P=T | context) should be 1. In a vacuum, if we are not given this premise, asking whether it is true that Tom reads Game Theory seems a strange question, since we do not know anything about Tom. Can you define your concept of probability measure here, and if/how it is conditioned on some given assumed-to-be-true context?
6. Relatedly, your current definition of p(Q=T) at line 274 is defined recursively. If a context is provided such that the problem is well-defined in a propositional sense (i.e. Q is deducible by propositional logic), then presumably the recursive computation would get to a point where the probability of the atomic premises E would be either 0 or 1, and then this would lead to p(Q=T) eventually being either 0 or 1. This is all hypothetical, however, since the base measure of p(Q=T) is not defined, and I'm assuming it would be defined such that p(Q=T | Q=T) = 1. When you prompt the LLM for p(E), do you provide it the context of the problem? Typically, problems include some single-variable clauses (in CNF form) which define atomic truths with which one can then deduce the query using the provided multi-variable clauses (inference rules).
7. Relatedly, given your definition of P(Q=T), which is computed over every proposition in the problem, are logically irrelevant propositions which were included in the problem not included in your computation? If my problem is "John likes candy. If John likes candy he is young. Sarah hates candy. Q: John is young", would your computation not include P(Sarah hates candy)?
8. Relatedly, given your definition of P(Q=T), is it not true that P(Q=T) = P(\neg Q = T), since the premises E of the question do not change with the query? If your computation of each atomic probability during the recursion is based on the contextual problem, then this would not be true as eventually you would have that p(Q =T | \neg Q = T) = 0 instead of P(Q = T | Q = T) = 1. Again, this would demand some notion of conditioning on the given problem context, rather than assuming independence across premises/propositions.
9. You mention that CNF conversion of logic resulted in "exponential explosion" (l. 217). I find this surprising, since typically problems are naturally described such that all listed premises must be simultaneously true. In cases where a single premise includes a conjunction in the antecedent, DeMorgans leads to a simple conversion to disjunction. In cases where a single premise includes a conjunction in the consequent, splitting the premise into two premises for each variable in the conjunction works fine. Can you explain further how this explosion occurs?
10. Experimental results: why do you use exact match (EM) for FOLIO? To my knowledge, it is a multiple choice dataset where the choices are True, False, or Unknown. Why do you use acc_norm for LogiQA and MathQA? It seems that these datasets are labelled with a single correct answer. If they don't, should you not use some sort of multi-label classification metric, like Hamming distance or Acc@K?
11. Experimental results: to what degree does your method improve over non-fine-tuned approaches, such as single in-context prompting and self-consistency? To what degree does your method improve over alternative curriculum schemes, such as the ones described in your related works section?
12. In general, can you please link to specific appendices when referencing them in the main text, rather than a general "See the supplementary material [...]"?
13. l. 255: "The syntactic complexity of each DNF formula [...] is entirely determined by a sequence of integers". This claim is neither substantiated in the paper, nor true. If this were true, two problems with identical sequences would be identically complex to solve. A vast literature on the measurement of empirical solution complexity of propositional logic problems exists, demonstrating that complexity is not this simple. For instance, see [1]
14. While not necessary to the paper, an exploration of empirical measurements of propositional complexity via SAT-solver-based metrics (# conflicts, time-to-solve, tree-depth) would be interesting to see.


[1]  Ansótegui, Carlos, et al. "Measuring the Hardness of SAT Instances." AAAI. Vol. 8. 2008.

---

### Author Response · Authors · 2025-12-02

We would like to express our sincere gratitude to all reviewers for the constructive reviews that are helpful to improve our work. We will revise our manuscript based on them.

---

### Meta-Review · Area_Chair_xZtn · 2025-12-16

**Summary:**

Reviewers like the high-level idea of phased fine-tuning (easy to hard) with an automatic difficulty metric combining DNF-based structure and an LLM-based semantic component. The decision-critical concerns are (i) shaky/unclear formalism around probabilistic framing and “independence,” plus some unsupported claims, (ii) weak experimental framing with sparse baselines and unclear metric choices, and (iii) missing implementation details (especially how proposition probabilities are extracted and the lack of code). On top of that, there are scope concerns: the pipeline is tightly tied to propositional logic and relies on NL-to-symbolic translation whose errors may propagate, and the method seems to need dataset/model-specific tuning.

**Reviewer Concerns:**

No rebuttal.

**Reviewer Scores:**

NA

---

### Decision · Program_Chairs · 2026-01-26

Reject